# LSD1 Inhibition Enhances the Immunogenicity of Mesenchymal Stromal Cells by Eliciting a dsRNA Stress Response

**DOI:** 10.3390/cells11111816

**Published:** 2022-06-01

**Authors:** Fatemeh Mardani, Wael Saad, Nehme El-Hachem, Jean-Pierre Bikorimana, Mazen Kurdi, Riam Shammaa, Sebastien Talbot, Moutih Rafei

**Affiliations:** 1Department of Pharmacology and Physiology, Université de Montréal, Montréal, QC H3T 1J4, Canada; fatemeh.mardani@umontreal.ca (F.M.); wael.saad@umontreal.ca (W.S.); hachemn@gmail.com (N.E.-H.); sebastien.talbot@umontreal.ca (S.T.); 2Department of Chemistry and Biochemistry, Lebanese University, Hadat, Lebanon; mazen_kurdi@hotmail.com; 3Pediatric Hematology-Oncology Division, Centre Hospitalier Universitaire Sainte-Justine Research, Montréal, QC H3T 1C5, Canada; 4Department of Microbiology, Infectious Diseases and Immunology, Université de Montréal, Montréal, QC H3T 1J4, Canada; jean.pierre.bikorimana@umontreal.ca; 5Intellistem Technologies Inc., Toronto, ON M5R 3N5, Canada; rshammaa@intellistemtech.com; 6Molecular Biology Program, Université de Montréal, Montréal, QC H3T 1J4, Canada

**Keywords:** mesenchymal stromal cells, antigen presentation, LSD1, tranylcypromine, UM171a, dsRNA, MHCI peptides, T-cell lymphoma, cellular vaccine, anti-tumoral response

## Abstract

Mesenchymal stromal cells (MSCs) are commonly known for their immune-suppressive abilities. However, our group provided evidence that it is possible to convert MSCs into potent antigen presenting cells (APCs) using either genetic engineering or pharmacological means. Given the capacity of UM171a to trigger APC-like function in MSCs, and the recent finding that this drug may modulate the epigenome by inhibiting the lysine-specific demethylase 1 (LSD1), we explored whether the direct pharmacological inhibition of LSD1 could instill APC-like functions in MSCs akin to UM171a. The treatment of MSCs with the LSD1 inhibitor tranylcypromine (TC) elicits a double-stranded (ds)RNA stress response along with its associated responsive elements, including pattern recognition receptors (PRRs), Type-I interferon (IFN), and IFN-stimulated genes (ISGs). The net outcome culminates in the enhanced expression of H2-K^b^, and an increased stability of the cell surface peptide: MHCI complexes. As a result, TC-treated MSCs stimulate CD8 T-cell activation efficiently, and elicit potent anti-tumoral responses against the EG.7 T-cell lymphoma in the context of prophylactic vaccination. Altogether, our findings reveal a new pharmacological protocol whereby targeting LSD1 in MSCs elicits APC-like capabilities that could be easily exploited in the design of future MSC-based anti-cancer vaccines.

## 1. Introduction

Mesenchymal stromal cells (MSCs) were long perceived to be potent immune-suppressive cells mediating their effects via two distinct mechanisms: (i) the secretion of paracrine-acting soluble factors, and/or (ii) the activation/reprogramming of endogenous suppressive myeloid cells following MSC uptake by efferocytosis [1,2,3,4]. MSCs can, however, be reprogrammed to behave as antigen presenting cells (APCs) that are capable of capturing and cross-presenting extracellular antigens [5,6]. These new functions can be instilled through the de novo expression of specific proteasomal subunits, or in response to pharmacological stimulation [5,6,7,8]. The latter strategy is particularly interesting as it entails the use of small molecules other than interferon (IFN)γ to elicit a selective rather than wider activation of gene expression [5,7,8]. An example of such an approach uses UM171a, a pyrimido-indole derivative that was originally identified for its ability to ex vivo stimulate hematopoietic stem cell (HSC) expansion [5,9,10,11,12,13,14,15]. The idea of treating MSCs with UM171a stems from the ability of this chemotype to trigger the expression of the co-stimulatory molecule CD86 on the surface of UM171a-treated HSCs [14]. Although the compound failed at eliciting CD86 expression on MSCs, UM171a treatment led to the up-regulation of major histocompatibility complex (MHC)I molecules, the enhanced production of reactive oxygen species (ROS), and the de novo expression of the *Psmb8* immunoprotrateasome subunits [5]. As a result, UM171a-treated MSCs acquired the ability to cross-present soluble antigens and to mount an effective anti-tumoral response in the context of therapeutic vaccination, using a pre-clinical T-cell lymphoma model [5]. This proof-of-concept study not only demonstrates the versatile framework of MSC pharmacological treatments to modulate their innate function, but also provides the impetus to establish a new line of study aimed at converting these highly sought cellular biopharmaceuticals into an anti-cancer vaccination tool.

Several factors highlight the importance of targeting lysine-specific demethylase 1 (LSD1) as a means for stimulating APC-like functions in MSCs. To start, a recent study revealed how UM171 targets the degradation of the histone H3K4 demethylase LSD1 [14,15]. The same study provided convincing data that LSD1 inhibition mimics the effect of UM171a on HSC expansion [15]. Consistent with this observation, the initial screen was used to identify UM171a unveiled tranylcypromine hydrochloride (an LSD1 inhibitor) as the second-best hit after UM171a for HSC expansion [13]. Additionally, this H3K4 demethylase plays critical roles in suppressing endogenous double-stranded (ds)RNA levels, which are crucial elements in driving the expression of pro-inflammatory type-I IFNs [16]. We thus reasoned that the pharmacological inhibition of LSD1 in MSCs may represent a plausible reprograming strategy for conveying antigen presentation/cross-presentation functions in MSCs. Such a study will not only unveil whether new pharmacological compounds can convert MSCs into APCs, but it will also provide fundamental knowledge regarding the mode of action of these two compounds with respect to targeting the entire CoREST complex (for UM171a), as opposed to a specific component within this complex (specific inhibitor of LSD1). In a nutshell, this is the first study demonstrating that tranylcypromine (TC)-mediated LSD1 inhibition in MSCs induces UM171a-distinct APC-like functions through a dsRNA stress response, consequently resulting in efficient and stable peptide presentation to responding CD8 T cells and potent anti-tumoral effector responses.

## 2. Materials and Methods

### 2.1. Animals and Ethics

All C57BL/6 female mice used in this study were 6–8 weeks old and purchased from Jackson Laboratories (Bar Harbor, ME, USA). The mice were housed in a pathogen-free environment at the animal facility of the Institute for Research in Immunology and Cancer. The Animal Ethics Committee of Université de Montréal approved all experimental procedures used.

### 2.2. Cell Lines and Reagents

The B3Z and E.G7 T-cell lymphoma lines were kindly provided by Dr. Jacques Galipeau (University of Wisconsin–Madison, WI, USA). The human umbilical cord (UC)-derived MSCs and their corresponding culture media were purchased from RoosterBio (Frederick, MD, USA). The cell culture media used to grow MSCs, EG.7, and B3Z were purchased from Wisent Bioproducts (St Jean-Baptiste, Montréal, QC, Canada). The UM171a compound was provided by ExCellThera (Montreal, QC, Canada). Albumin from chicken egg white (OVA), Accutase^®^, (TC), Chlorophenol red-β-D-galactopyranoside (CPRG), and Amicon Ultra-4 centrifugal filters were purchased from Sigma-Aldrich (St-Louis, MI, USA). Antibodies used in the flow cytometry analysis were purchased from BD Biosciences (San Jose, CA, USA). The anti-dsRNA antibody (J2 clone) was purchased from Scicons (Susteren, The Netherlands). The indoleamine 2,3-dioxygenase (IDO)-1 ELISA was purchased from Cusabio Technology LLC (Houston, TX, USA). Murine IFNβ and human IFNγ quantikines, as well as the anti-endothelial protein C receptor (EPCR) antibody, were purchased from RnD systems (Toronto, ON, Canada). Recombinant murine IFNγ was purchased from Peprotech (Rocky Hill, NJ, USA). The SIINFEKL (OVA 257–264) and YMLDLQPET (HPV E7 11–19) peptides were synthesized by Genscript (Piscataway, NJ, USA). The annexin-V staining kit was purchased from Cedarlane (Burlington, ON, Canada).

### 2.3. Bioinformatics Analysis

We downloaded TC perturbational data from GSE34672. The corresponding gene expression data was normalized and analyzed through the GEO to R interface. Differential gene expression ratios were derived from the comparisons of TC to untreated TEX cell lines (transformed to mimic undifferentiated primary leukemia cells). A selection of the best representative samples in each of the treated to untreated groups was based on the first principal component. Gene set enrichment analysis was performed on genes ranked by the corresponding log2 fold change. The ggplot2, heat-map, and custom scripts through the R programming language were used to create the analytical framework and figures.

### 2.4. Generation of Primary Bone Marrow (BM)-Derived MSCs

To generate mouse primary MSCs, whole BM was flushed of femur bones derived from a C57BL/6 mouse, and cultured in AMEM supplemented with 10% fetal bovine serum (FBS) and 50 µM/mL penicillin–streptomycin, as previously reported [8]. The media was changed every 2–3 days until MSC colonies were apparent. Following 2–3 passages and the appearance of a uniform population, MSCs were phenotyped using flow-cytometry using antibodies against CD44, CD45, CD73, CD90.1, CD105, and H2-K^b^ diluted according to the manufacturer’s instructions. The samples were acquired by BD FACS Diva on CANTOII, then analyzed using FlowJoV10.

### 2.5. Identification of the TC Maximum Tolerated Dose (MTD) and Subsequent Treatments

To identify the MTD for TC, 3 × 10^4^ MSCs were plated in a 6-well plate, then treated with ascending doses of the compound (100 to 2000 µM) for 24 h. Treated cells were then detached using trypsin prior to trypan blue staining and counting live and dead cells. The dose with no toxic effects (e.g., no cell death) or decreased proliferation was selected for all subsequent studies. A similar approach was used for human UC-MSCs. The identified dose of 500 μM (and 1000 μM for human UC-MSCs) was then used in all subsequent experiments over an incubation period of 48 h. For the EPCR comparative analysis, UM171a was used at 1000 nM for 72 h as previously reported [5].

### 2.6. RNA Extraction

To quantify gene expression, 1.5 × 10^5^ cells were plated in a 10 cm petri dish, then treated with 500 μM TC or an equivalent volume of DMSO for 48 h. A total of 5 × 10^5^ cells were then collected for RNA extraction using the Qiagen RNeasy kit (n = 6/group). The RNA quality was assessed using a quick-drop spectrophotometer. Samples were stored at −80 °C until qPCR analysis.

### 2.7. Assessment of dsRNA via Confocal Micropscopy

For confocal microscopy, 1.5 × 10^4^ TC- or DMSO-treated MSCs were plated on gelatin-coated cover slides in a 24-well plate for 2 days. Following treatments, cells were fixed with 2% paraformaldehyde (500 μL/well) for 15 min prior to their washing 3× with 500 uL of PBS and permeabilization using 0.2% TX-100 (500 μL/well) for 5 min. The cells were then washed with 0.1% BSA (500 μL/well) then incubated with 1% BSA solution. One hour later, cells were washed 3× with 0.1% BSA (500 μL/well), then stained using the mouse anti-dsRNA antibody (200 μL/well) for 1 hr. After washing, a secondary goat anti-mouse IgG-AF488 antibody was added at 1:100 for 1 h, followed by washing with 0.1% BSA (500 μL/well). The slides were then mounted and kept overnight to dry prior to their analysis using the LSM 800 confocal laser-scanning microscope (Carl Zeiss, Montréal, QC, Canada).

### 2.8. Antigen Presentation Assay

To assess antigen presentation, MSCs were plated in a 24-well plate at 5-, 10-, 15-, or 20 × 10^4^ cells per well for 7-, 3-, 2-, and 1-day treatments, respectively. Following plating, MSCs were treated with 500 μM of TC or an equivalent volume of DMSO for a relative duration. Once the treatments were completed, cells were washed with PBS and then pulsed with the OVA protein at 5 mg/mL for 6–7 h, or the SIINFEKL peptide at 1 μg/mL for 2–3 h. Following the incubation period, pulsed cells were washed with PBS to remove excess antigen/peptide, followed by the addition of 5 × 10^5^ B3Z cells (SIINFEKL/H2-K^b^-specific T-cell line) for 15–17 h. The following day, all cells were lysed using the RIPA buffer, then stained with a CPRG solution (2.3 mg in 17 mL buffer) for 18 h at 37 °C. The optical density signal was measured using a microplate reader at a wavelength of 570 nm.

### 2.9. Cytokine Quantification

To quantify cytokine production, 10^6^ MSCs treated with 500 μM of TC- or an equivalent volume of DMSO were cultured for two days in serum-free media. Once the incubation period was completed, the conditioned media were collected and centrifuged for 10 min at 800× *g* to remove any floating cells or debris prior to their concentration, using the Amicon Ultra-4 centrifugal filters (3000 NMWL) for 45 min at 4500× *g* at 4 °C. Collected concentrates were then frozen at −80 °C until they were shipped to EveTechnologies (Calgary, AB, Canada) for cytokines analysis by Luminex. The quantification of IFNβ was conducted using a commercial quantikine according to the manufacturer’s instructions.

### 2.10. Monitoring Peptide: MHC Complex Stability

To evaluate the effect of TC on antigen presentation stability, 1.5 × 10^4^ cells/well were seeded in a 24-well plate. On the following day, cells were treated with TC (500 μM) or an equivalent DMSO volume for 2 days. Once the TC treatment period was completed, 1 μg/mL of the SIINFEKL peptide was used to pulse MSCs for 12-, 6-, 4-, and 2 h. Following the incubation period, cells were detached using Accutase (to preserve the MHCI: peptide complex), then stained with the SIINFEKL:H2-K^b^ complex-specific 25-D1.16 antibody for 30 min. Cells were then washed twice with cold PBS containing 2% FBS prior to their analysis, using flow-cytometry.

### 2.11. Assessing ROS Production in Response to TC Treatment

TC- or DMSO-treated MSCs (n = 3) were collected and washed with PBS containing 2% FBS. Mitochondrial ROS production was measured by re-suspending 2 × 10^5^ cells in 100 μL of 5 μM MitoSox staining for 15 min at 37 °C in the dark [5]. Once the incubation period was completed, cells were washed and resuspended in 2% FBS in PBS. Analysis was then assessed using flow cytometry. A similar approach was used to investigate cytoplasmic ROS, except that cells were stained in 10 μM DHE for 30 min at 37 °C in the dark, according to manufacturer’s protocol.

### 2.12. Treatment with IFNγ and IDO-1 Quantification

One million MSCs were treated in 500 μM of TC, 50 ng/mL of IFNγ, or an equivalent volume of DMSO for 48 h in serum-free media. Once the incubation period was completed, conditioned media were collected and centrifuged for 10 min at 800× *g* to remove any floating cells or debris prior to concentrating the collected mediam, using the Amicon Ultra-4 centrifugal filters (3000 NMWL) for 30 min at 4500× *g* at 4 °C. The collected concentrate was then frozen at −80 °C until it was used for IDO-1 quantification using a commercial ELISA.

### 2.13. Design of the Prophylactic Vaccination Study

To test the protective potency of the vaccine, 6–8-week-old female C57BL/6 mice (n *= 5*/group) were subcutaneously (SC)-immunized at day 0 and 14 using 2.5 × 10^5^ C57BL/6-derived TC- or DMSO-treated MSCs pulsed for 3 h with the SIINFEKL peptide at 1 μg/mL (unpulsed TC- and DMSO-treated MSCs were used as controls). The mice were then challenged with 5 × 10^5^ EG.7 tumor cells via SC injection, at one week following the second dosing. Tumor volumes were then assessed over 30 days or until reaching set endpoints (e.g., ulceration or tumor volumes ≥ 1000 mm^3^).

### 2.14. Statistical Analysis

*p*-values were calculated accordingly, using either the student *t* test or the one-way analysis of variance (ANOVA). Results are represented as the average mean with standard deviation (S.D.) error bars, and statistical significance is represented with asterisks: * *p* ˂ 0.05, ** *p* ˂ 0.01, *** *p* ˂ 0.001.

## 3. Results

### 3.1. TC-Treated MSCs Exhibit a Significant Increase in H2-K^b^ Expression

The epigenetic landscape is complex and it encompasses a large combination of molecular events [16]. Amongst the diverse key determinants controlling transcriptional regulation is LSD1 [17]. This flavin-containing amino oxidase is capable of specifically demethylating mono- or di-methylated lysine 4 at histone H3 (H3K4me1 and H3K4me2) in a flavin adenine dinucleotide-monoamine oxidase-dependent mechanism [17]. As such, LSD1 can be inhibited by monoamine oxidase-targeting agents such as TC [17]. In order to maximize the impact of TC treatment, we first conducted an MTD assay to identify a non-toxic dose with a limited impact on MSC proliferation. Thus, 4 × 10^4^ MSCs were plated one day prior to adding ascending TC doses (100–2000 μM; the DMSO dose used in this experiment was equivalent to the amount included in the 2000 μM dose). Overall, MSCs continued proliferating in the following 24 h, with 100, 200, and 500 μM of TC, whereas inhibitory effects were observed at higher doses (Figure 1A). We next assessed whether the TC treatment of MSCs modulates their innate phenotype, using flow-cytometry. Although both the TC- and DMSO-treated MSCs exhibited no apparent changes in their innate phenotype and remained CD44^+^, CD45^-^, CD73^+^, CD90.1^+^, and CD105^+^ (Figure 1B), the cell surface expression level of H2-K^B^ (MHCI molecule) was significantly up-regulated in the TC treatment group (Figure 1B,C). Despite a similar H2-K^b^ response to UM171a stimulation, TC-treated MSCs did not exhibit a higher EPCR expression, which is a major hallmark for UM171a responsiveness (Figure 1D). Thus, MSC treatment with TC enhances H2-K^b^ cell surface expression with no EPCR-induced effect, suggesting a UM171a-divergent mode of action [5].

### 3.2. Bioinformatics Analysis Predicts Induced Type-I IFN Signaling and Antigen Presentation in Response to TC Treatment

Responses to epigenetic drug treatments may be complex, given the wide scope of action for this drug class on the cellular genome. We thus elected to first analyze the publicly available transcriptomic data generated from human TEX cells treated with TC as a means to predict the potential reactome processes that may be relevant to our main objective [18]. Our investigation shows that the TC treatment of TEX cells induces various immune-related pathways (Figure 2A). Among these, type-I IFN signaling, as well as antigen processing and presentation, were significantly up-regulated (FDR < 0.05 from GSEA; Figure 2B). We further analyzed specific genes in these processes and observed an interesting increase in the expression of MHCI (HLA-A and HLA-H), IFNα, and β signaling

(*Ifi6*, *Ifi35*, *Ifit1*, *Oas3*, *Irf1*, *Irf9*, *Bst2*, *Jak1*, *Isg15*, *Ifnar2*, *Ifnar1*, and *Ptpn1*), as well as immunoproteasome subunits (*Psmb8*, *Psmb9* and *Psmb10*-Figure 2C). The significant modulation of these genes, along with other genes related to these two main immune-related reactomes, clearly suggests that TC treatment may affect type-I IFN signaling and antigen processing and presentation.

### 3.3. MSCs Express dsRNA and Their Cognate Sensing Elements in Response to TC Treatment

In addition to the observed increase in MHCI (H2-K^b^) expression, the LSD1 complex is known to repress the expression of endogenous retroviral viruses (ERVs), as well as several genes that are involved in sensing dsRNA expression, such as PRRs, Type-I IFN, and ISGs [16]. We thus treated MSCs with 500 μM of TC and confirmed the drug’s ability to enhance the expression of dsRNA using confocal microscopy (Figure 3A,B). We next quantified the transcript levels of various antigen presentation- and dsRNA response-related genes [16]. No changes were detected in the expression of *Tap1*, *Tapbp*, or *Erap1*, whereas a slight but significant decrease was observed for *Tap2*, *Calr*, and *Pdia3* (Figure 3C). Interestingly, *B2m* expression was increased in response to TC treatment (Figure 3C), which is consistent with the enhanced H2-K^b^ levels previously detected at the cell surface of TC-treated MSCs (Figure 2B,C). The quantification of the two main Type-I IFN genes, on the other hand, revealed a robust expression of IFNβ with no detectable signal for IFNα (Figure 3D). In parallel, the analysis of PRR-, ISG-, and interferon-responsive element (IRFs)-related genes show potent increases in *Oasl2* and *Isg15* (Figure 2C), as well as in *Irf7* (Figure 3D). Since we anticipate TC to behave in a manner that is similar to UM171a, we next quantified the expression level of both the constitutive (*Psmb5*, *Psmb6*, and *Psmb7*) and immunoproteasome (*Psmb8*, *Psmb9*, and *Psmb10*) subunits. Interestingly, TC treatment triggered no major fluctuations in the expression of constitutive proteasome genes, whereas *Psmb8* and *Psmb9* expression (immunoproteasome subunits) was enhanced compared to the DMSO treatment group (Figure 2G). Altogether, these data demonstrate that TC treatment triggers the expression of dsRNA and related response elements, including the components of the immunoproteasome complex, which may have an important impact on MSC immunogenicity.

### 3.4. TC-Mediated LSD1 Inhibition Improves MSC Antigen Presentation through Cell Surface MHC:Peptide Complex Stabilization

Given the observed effects of TC on H2-K^b^, ERVs, and the immunoproteasome, we reasoned that the antigen presentation/cross-presentation ability of TC-treated MSCs may be improved. We thus conducted several antigen presentation (using the SIINFEKL peptide) and cross-presentation (using the full OVA protein) assays at different time points (Figure 4A). Although the SIINFEKL pulsing of DMSO-treated MSCs triggers comparable B3Z activation (gray bars) at all tested timelines (days 1, 2, 3, and 7), treatment with the OVA protein had no detectable signal (blue bars, Figure 4B). Interestingly, SIINFEKL pulsing of TC-treated MSCs enhanced the T-cell response in the first 48 h before returning to basal levels at days 3 and 7 (green bars-Figure 4B). In contrast, there was no detectable T-cell activation in response to OVA protein pulsing (red bars-Figure 4B). Since MSCs are known to modulate target cells in a paracrine fashion via the secretion of soluble mediators, we next investigated whether TC-induced epigenetic changes could cause a noticeable change in both cytokine (Figure 4C) and chemokine (Figure 4D) production. Although slight changes were observed for VEGF, IL-6, IP-10, LIF, and RANTES production, IFNβ was the only cytokine to be strongly induced in response to TC treatment (Figure 4C). Since the combined increase in β2m and IFNβ expression could account for the enhanced MSC immunogenicity (B3Z response observed earlier) by directly affecting MHCI, we next investigated the stability of the peptide: MHCI complex following TC treatment. SIINFEKL pulsing of DMSO-treated MSCs leads to complex formation 3 h post-pulsing, prior to its disappearance 24 h later (Figure 4E-left panel). In contrast, TC treatment of MSCs stabilized the peptide: MHC complex throughout all tested timelines, in addition to being rapidly detected on the cell surface (within 30 min) post-SIINFEKL pulsing, compared to the DMSO group (Figure 4E-right panel). Consistent with the absence of antigen cross-presentation, the peptide: MHC stability could not be attributed to ROS production, as their levels in TC-treated MSCs were significantly lower than those of UM171a-treated MSCs (Figure 4F,G) [5]. Thus, TC enhances the antigen presentation capacity of MSCs by stabilizing cell surface peptide: MHCI complexes with no noticeable effect on antigen cross-presentation.

### 3.5. IFNγ Treatment Does Not Synergise with TC at Enhancing Antigen Presentation

IFNγ is one of the most potent pro-inflammatory cytokines [6]. Not only can it promote the development of T-helper 1 immune responses and the development of potent cytotoxic T cells, but it was also shown to re-program MSCs to behave as conditional APCs [6]. One of IFNγ-stimulated MSC hallmarks is the up-regulation of MHCI and its associated molecular machinery, consequently resulting in enhanced antigen presentation and cross-presentation [6]. Based on this notion, we sought to investigate whether LSD1 inhibition combined with IFNγ treatment could lead to additive or synergistic effects on the antigen presentation potential of MSCs. When first assessed for MHCI expression, both TC and IFNγ treatments led to their cognate expected outcome (Figure 5A). Interestingly, their combined treatment did not lead to additive or synergistic effects on the MHCI intensities of MSCs in response to IFNγ or IFNγ when combined with TC, as the values were comparable (Figure 5A). In an attempt to further confirm this observation, we quantified IDO-1 levels (normally induced in response IFNγ treatment) in the supernatants of treated MSCs [5]. Compared to the IFNγ group, TC-treated MSCs had no detectable IDO-1 (Figure 5B), whereas the combination of TC and IFNγ impaired IDO-1 secretion significantly, clearly indicating an antagonist effect mediated via LSD1 inhibition (Figure 5B). Nevertheless, when both of these treatments were evaluated in the context of an antigen presentation assay, TC treatment enhanced the T-cell response, as expected, and it was further amplified via the use of IFNγ (Figure 4C). However, antigen presentation and cross-presentation by MSCs treated with a combination of TC and IFNγ were no different from the IFNγ treatment group alone (Figure 5C). These data indicate that TC-mediated LSD1 inhibition does not synergize with IFNγ in antigen presentation or cross-presentation, but it may impair some of IFNγ-triggered effects, as shown by the expression of the IDO-1 immune-suppressive factor.

### 3.6. TC-Mediated LSD1 Inhibition Enhances the Anti-Tumoral Property of MSCs

Given the enhanced in vitro antigen presentation ability triggered by TC, we next assessed the ability of SIINFEKL-pulsed TC-treated MSCs to elicit an anti-tumoral immune response in immunocompetent animals (Figure 6A,B). Using the EG.7 T-cell lymphoma model, we designed a prophylactic vaccine protocol in which naïve C57BL/6 mice (H2^b^) were first subcutaneously-immunized at days 0 and 14 with a low dose of syngeneic (H2^b^) MSCs (2.5 × 10^5^ cells). One week following the second dosing, the animals were challenged with 5 × 10^5^ EG.7 tumor cells (Figure 6B). In contrast to mice vaccinated with unpulsed DMSO (gray line), TC-treated (blue line) unpulsed MSCs delayed tumor growth significantly compared to unvaccinated animals challenged with EG.7 tumor (black line-Figure 6C). On the other hand, SIINFEKL-pulsed TC-treated MSCs (red line) exhibited a potent anti-tumoral response that was superior to SIINFEKL-pulsed DMSO-treated MSCs (green line-Figure 6C), with an overall survival rate of 80% (Figure 6D). The anti-tumoral response obtained with SINNFEKL-pulsed TC-treated MSCs correlated with the significant increase in the levels of peripheral CD8 T_Eff_ (7.36 ± 1.91%) and T_CM_ (14.46 ± 2.64%), with no detectable difference being observed in the CD4 compartment Figure 6E,F). Of note was the significant increase in the percentage of CD8 T_Eff_ in the TC-treated MSC group, despite the absence of SIINFEKL pulsing (Figure 6F). Altogether, these findings are consistent with the in vitro antigen presentation data and provide clear evidence that LSD1 inhibition in MSCs can be exploited for the design of an anti-cancer cellular vaccine.

### 3.7. TC Treatment of Human-Derived UC-MSCs Triggers Similar Effects

To ensure that the effects obtained on murine MSCs can be replicated using human cells, we next conducted a TC MTD study on human UC-derived MSCs. As shown in Figure 7A, the treatment of UC-MSCs with several TC concentrations revealed that the 500 and 1000 μM doses are both well-tolerated. When further tested on UC-MSCs, TC treatment enhanced HLA-ABC expression compared to DMSO treatment (Figure 7B). We next conducted an antigen presentation assay using an E7 HPV-derived HLA.A2-specific peptide (Figure 7C) and found that TC-treatment of UC-MSCs enhanced HPV-specific CD8 T-cell activation (Figure 7D). In addition, TC treatment of UC-MSC led to the up-regulation of several genes (*Oasl2*, *Ifih1*, *Dhx58*, *Tlr3*, and *Psmb8*) previously observed in TC-treated murine cells (Figure 7E). Altogether, these results indicate that akin to murine MSCs, TC treatment is well-tolerated by human UC-MSCs and triggers an increase in HLA.ABC cell surface expression, along with enhanced antigen presentation properties.

## 4. Discussion

The fact that LSD1 ablation mimics the effects induced by UM171a prompt us to investigate whether the pharmacological inhibition of LSD1 activity can convert immune-suppressive MSCs into APCs [15]. As such, we demonstrate that the TC-based inhibition of LSD1 causes intracellular dsRNA stress, resulting in the de novo production of IFNβ and an increase in cell surface expression and the stability of MHCI (H2-K^b^) molecules. The net outcome culminates in the enhanced capacity of TC-treated MSCs to stably present immunogenic peptides to responding CD8 T cells, resulting in the induction of a potent anti-tumoral effector response (graphical abstract). A key difference between the TC and UM171 treatments is that the latter instills antigen cross-presentation capacities in MSCs (presenting peptides on MHCI from captured proteins), while the former acts mostly at the level of antigen presentation (ideal for peptide pulsing). This difference may be explained by two major factors. First, UM171a may have the ability to affect several epigenetic-related protein complexes, including LSD1, whereas TC may be more selective towards LSD1 within the CoREST complex [15]. Second, TC does not reproduce the exact effects triggered by UM171a, such as ROS production [15]. This is particularly important, as mitochondrial-derived ROS are sought to play crucial roles in driving antigen cross-presentation [5,7,8,19,20,21]. More specifically, ROS production protects endocytosed antigens from extensive degradation mediated via pH-sensitive proteases activated during endosome maturation [5,7,8,19,20,21]. Second, ROS can trigger lipid peroxidation, allowing for the release of captured antigens to the cytosol, where they can be efficiently processed by the proteasomal complex, and perhaps follow a sorting pathway that is specific to endosomal recycling to the cell surface [8,22]. This is consistent with previous observations whereby ROS neutralization completely blunts antigen cross-presentation and T-cell responses mediated via gene-engineered MSCs or UM171a-treated MSCs [5,8]. As such, ROS production may represent the limiting step required by TC-treated MSCs to cross-present exogenous antigens [5].

LSD1 inhibition in cancer cells has been also reported to stimulate anti-tumoral immunity [16]. More specifically, the ablation of this demethylase in B16 melanoma triggers ERV transcription, which is sensed by the PRRs TLR3 and MDA5 (the latter being enconced by the *Ifih1* gene), consequently eliciting type-I IFNs [16]. This explains the elevated expression of MHCI (H2-K^b^) detected on the surface of cancer cells, as well as enhanced tumor infiltration by endogenous cytotoxic T cells [16]. Although most of the molecular observations were recapitulated in TC-treated MSCs, our study highlights the importance of DNA methylation inhibitors in modulating the immunogenicity of MSCs. It must be noted, however, that the conversion of suppressive MSCs into APCs may not exclusively rely on dsRNA stress, but could rather depend on the activity of type-I IFNs. In support of the latter notion, IFNα-secreting MSCs (IFNα being downstream of the dsRNA response) were previously shown to exhibit dramatic inhibitory effects on tumor progression [23]. Mechanistically, this anti-tumoral property was attributed to IFNα-mediated inhibition of inducible nitric oxide (iNOS) expression in MSCs, which is a major suppressive element that is used by IFNγ- or TNF-alpha-stimulated MSCs to blunt T-cell responses [23]. Although we did not directly assess iNOS expression in TC-treated MSCs co-stimulated with IFNγ, we observed a significant decrease in IDO-1 secretion in response to a combined treatment with TC and IFNγ, which clearly suggests a TC-mediated antagonist effect on IFNγ. We also presume that the lack of complete inhibitory effects on IDO-1 expression is due to absent IFNα production by TC-treated MSCs, which only express IFNβ. Second, we observed that mice injected with TC-treated MSCs without SIINFEKL pulsing exhibit a significant delay in tumor growth. Since both MSCs and tumor cells were injected within the same area, IFNβ secretion by TC-stimulated MSCs may have primed local endogenous immune cells, therefore engendering a hostile environment for tumor establishment and growth. Additional mechanistic studies are needed to further elucidate the mechanisms by which type-I IFNs produced by TC-treated MSCs reverse the innate immunosuppressive property of MSCs, and whether these cells can indeed prime local immunity to reject cancer growth without antigen pulsing.

## 5. Conclusions

So far, we cannot claim with confidence that the effects exhibited by MSCs in response to TC-mediated LSD1 inhibition rely exclusively on gene demethylation. This would require the comparison of a larger panel of LSD1 inhibitors to type-I IFN treatments in order to identify common or divergent outcomes on both antigen presentation and cross-presentation. Nevertheless, this remarkable ability to convert immune-suppressive MSCs to APC-like cells via LSD1 inhibition may provide novel means for increasing the efficacy of any peptide-pulsed MSC-based vaccine that is aimed at stimulating anti-tumoral immunity.

## Figures and Tables

**Figure 1 cells-11-01816-f001:**
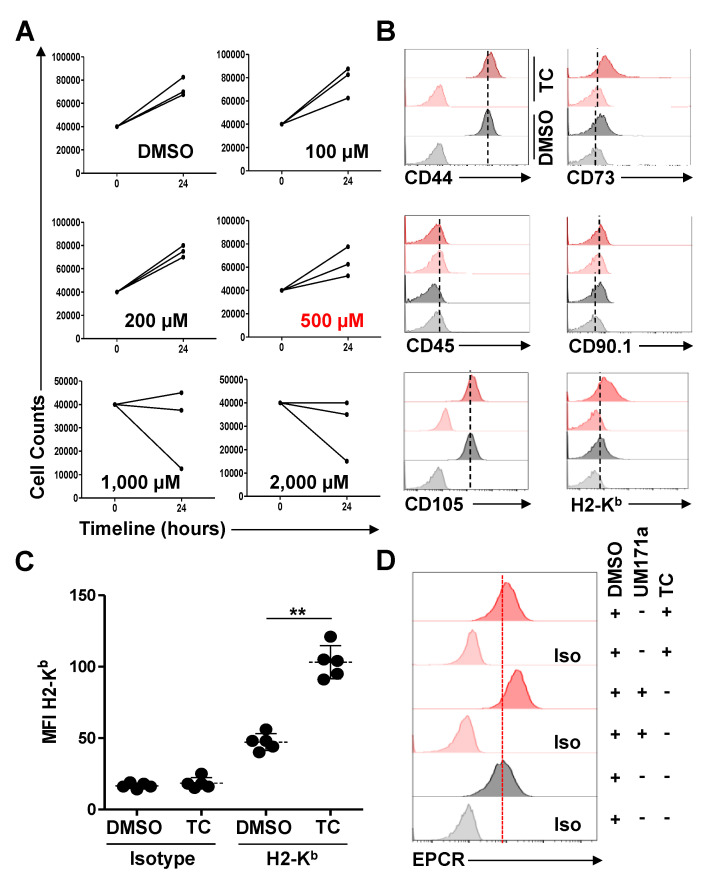
TC treatment enhances MHCI and EPCR expression on the surface of MSCs. (**A**) An MTD experiment conducted on MSCs using ascending doses of TC. Cell counting was conducted 24 h following treatment. For this experiment, n = 3/group. (**B**) Representative phenotypic analysis of MSCs treated with 500 μM of TC. DMSO-treated groups are shown in gray, whereas TC-related groups are displayed in red. (**C**) MFI of H2-K^b^ expression following DMSO or TC treatments. For this experiment, n = 5/group, with ** *p* < 0.01. (**D**) Representative flow-cytometry analysis of EPCR expression in response to TC and UM171a treatments.

**Figure 2 cells-11-01816-f002:**
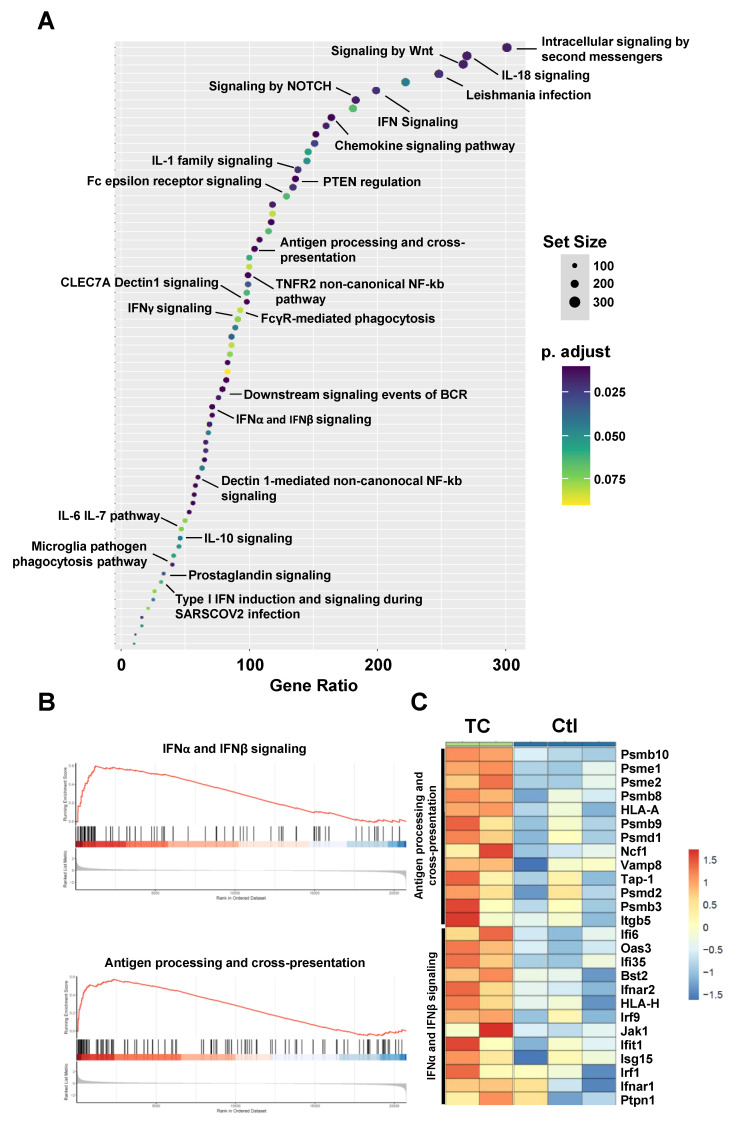
Bioinformatic analyses in TC-treated vs. untreated cells. (**A**) Dot plot graph showing top up-regulated reactome pathways at an FDR threshold < 10%. *X*-axis illustrates gene set ratio (keeping pathways with gene size between 10 and 500 for enrichment analysis). (**B**) Running-sum statistic for 2 reactome pathways: IFNα and IFNβ signaling (NES = 2; FDR = 0.1) and antigen processing and cross-presentation (NES = 2.05; FDR = 0.1). (**C**) Heat-map illustrating a set of genes that contributes most to the rank statistics from B) and significantly up-regulated using TC treatment.

**Figure 3 cells-11-01816-f003:**
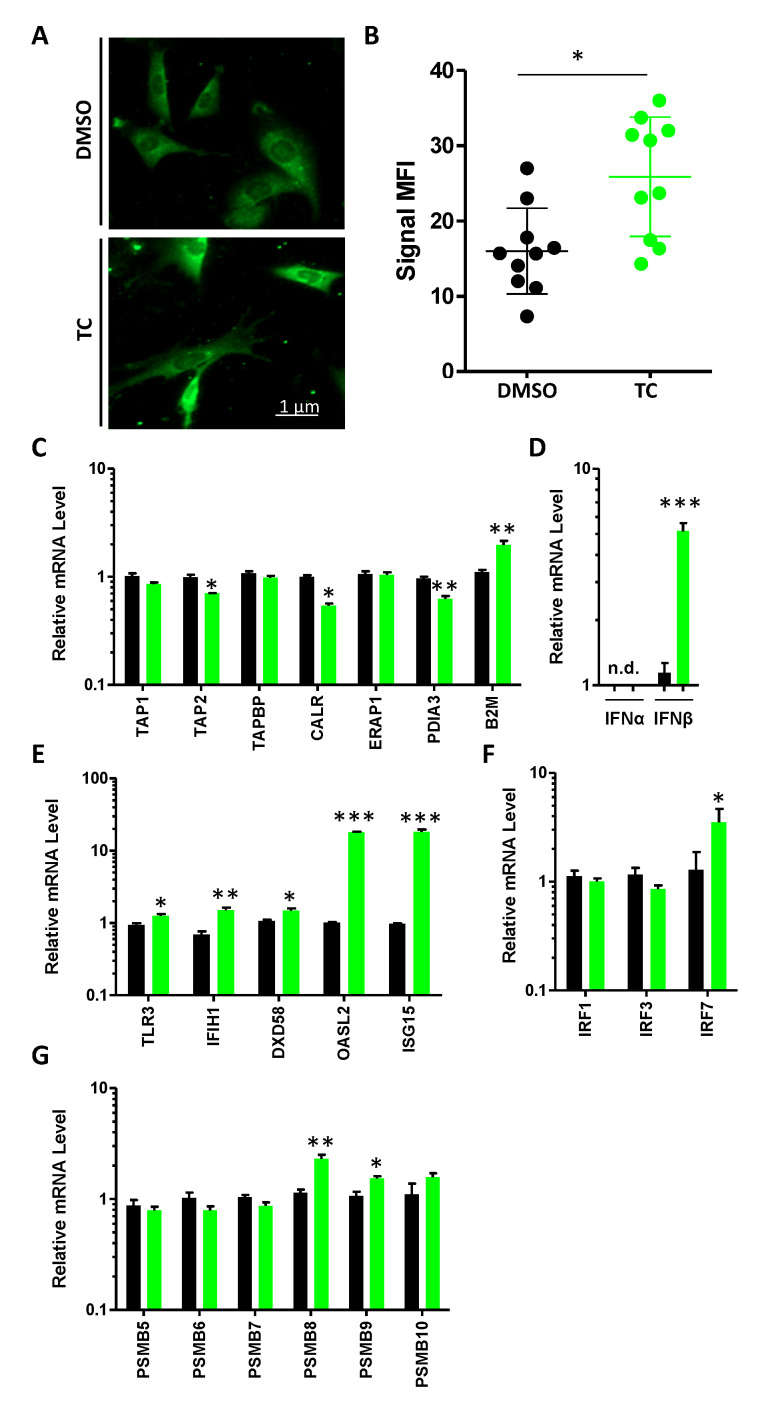
Characterizing the molecular response of TC-treated MSCs. (**A**,**B**) Representative confocal microscopy of dsRNA treatment in DMSO- or TC-treated MSCs, (**A**) and signal mean fluorescent intensity quantification (**B**). (**C**–**G**) Quantification of RNA transcripts for genes involved in antigen presentation (**C**), Type-I IFN (**D**), PRR (**E**), IRFs, (**F**) and proteasomal genes (**G**). For these experiments, n = 6/group with * *p* < 0.05, ** *p* < 0.01 and *** *p* < 0.001.

**Figure 4 cells-11-01816-f004:**
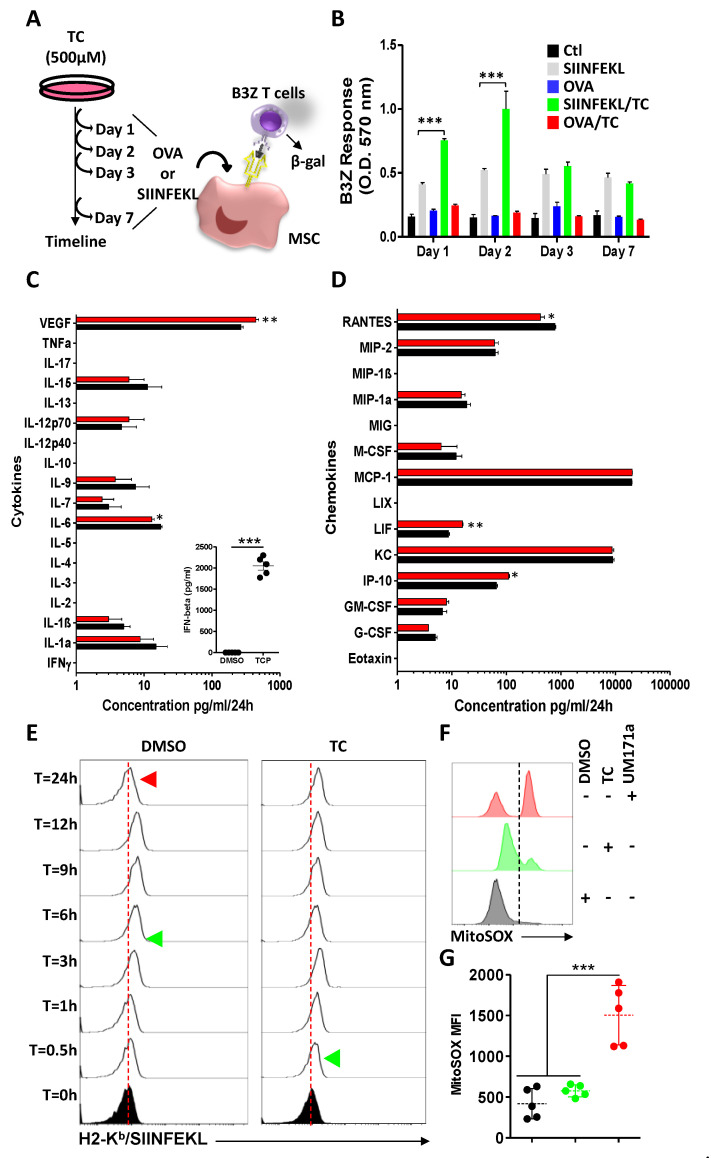
TC enhances the immunogenicity of MSCs. (**A**) Schematic diagram demonstrating the experimental design used for the antigen presentation assay. (**B**) B3Z activation in response to DMSO-treated MSCs (black), SIINFEKL-pulsed DMSO-treated MSCs (light gray), OVA-pulsed DMSO-treated MSCs (blue), SIINFEKL-pulsed TC-treated MSCs (green), and OVA-pulsed TC-treated MSCs (red). For this experiment, n = 6/group with *** *p* < 0.01. (**C**,**D**) Secretome analysis depicting cytokines (**C**), and chemokines (**D**) in response to DMSO (black) or TC (red) treatments. For these two experiments, n = 6/group with * *p* < 0.05 and ** *p* < 0.01. (**E**) Representative flow-cytometry experiments evaluating the stability of the SIINFEKL:H2-K^b^ complex over time in response to DMSO (left panel) and TC (right panel) treatments. (**F**,**G**) Representative flow-cytometry analysis assessing the ROS levels in the mitochondria (**F**) and its corresponding MFI (**G**) in response to DMSO (black), TC (green), or UM171a (red) treatments. The dotted line in panel F separates negative from positive ROS signals.

**Figure 5 cells-11-01816-f005:**
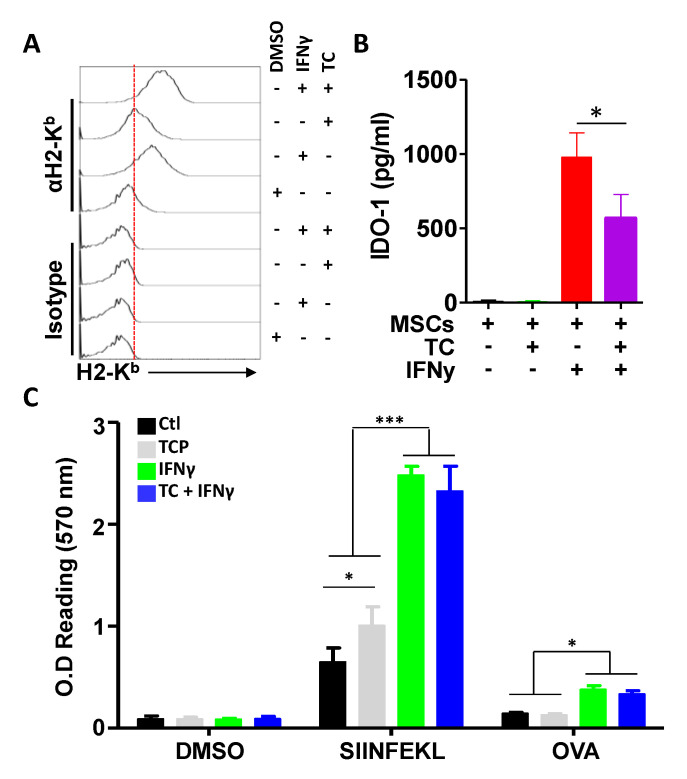
IFNγ and TC co-treatment does not trigger additive or synergistic effects. (**A**) Representative flow-cytometry experiment assessing the expression levels of H2-K^b^ in response to DMSO, IFNγ, or TC, alone or in combination with IFNγ. (**B**) IDO-1 quantification following IFNγ, TC, or combined treatments. (**C**) Antigen-presentation assay in response to IFNγ, TC, or combined compounds following SIINFEKL or OVA pulsing. For all panels shown in this figure, n = 6/group with * *p* < 0.05 and *** *p* < 0.001.

**Figure 6 cells-11-01816-f006:**
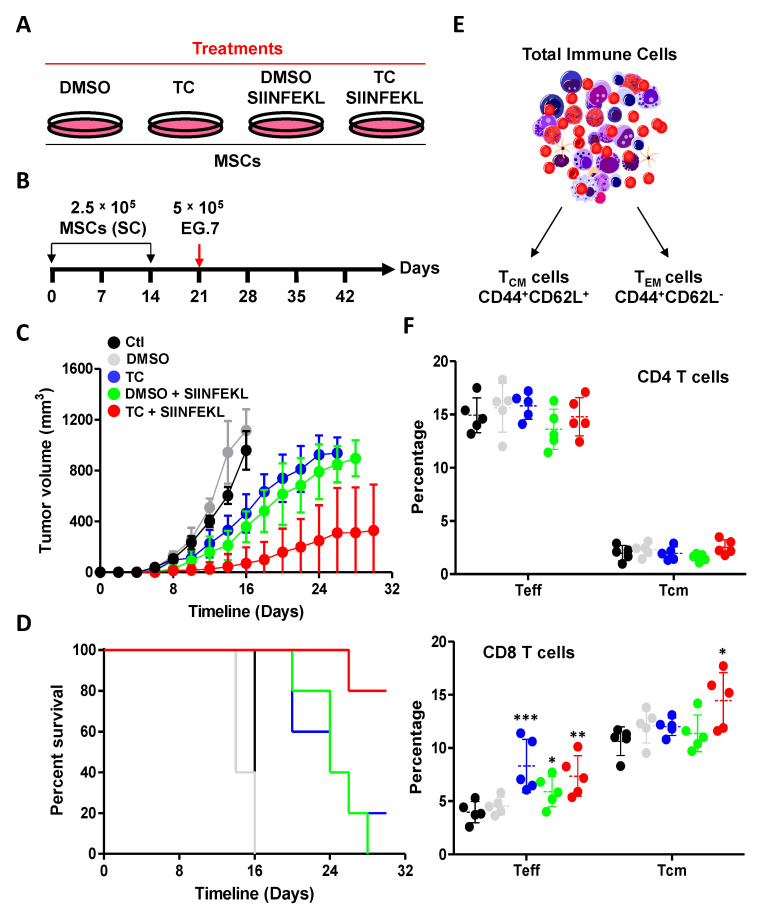
TC treatment enhances the immunogenicity of MSCs in the context of prophylactic vaccination. (**A**,**B**) Schematic diagrams showing the treatments groups (**A**) and experimental design (**B)** used in the vaccination protocol. (**C**) Tumor growth analysis over time. Control unvaccinated mice are shown in black, DMSO-treated MSCs in gray, TC-treated MSCs in blue, SIINFEKL-pulsed DMSO-treated MSCs in green, and SIINFEKL-pulsed TC-treated MSCs in red. (**D**) Kaplan–Meier survival curve for the experiment shown in panel (**C**). (**E**,**F**) Analysis of T_CM_, and T_Eff_ CD4 and CD8 T cells in vaccinated animals. Ctl mice are shown in black, DMSO in green, TC in red, DMSO + SIINFEKL in purple, and TC + SIINFEKL in blue. For this panel, n = 5/group with * *p* < 0.05, ** *p* < 0.01, and *** *p* < 0.001.

**Figure 7 cells-11-01816-f007:**
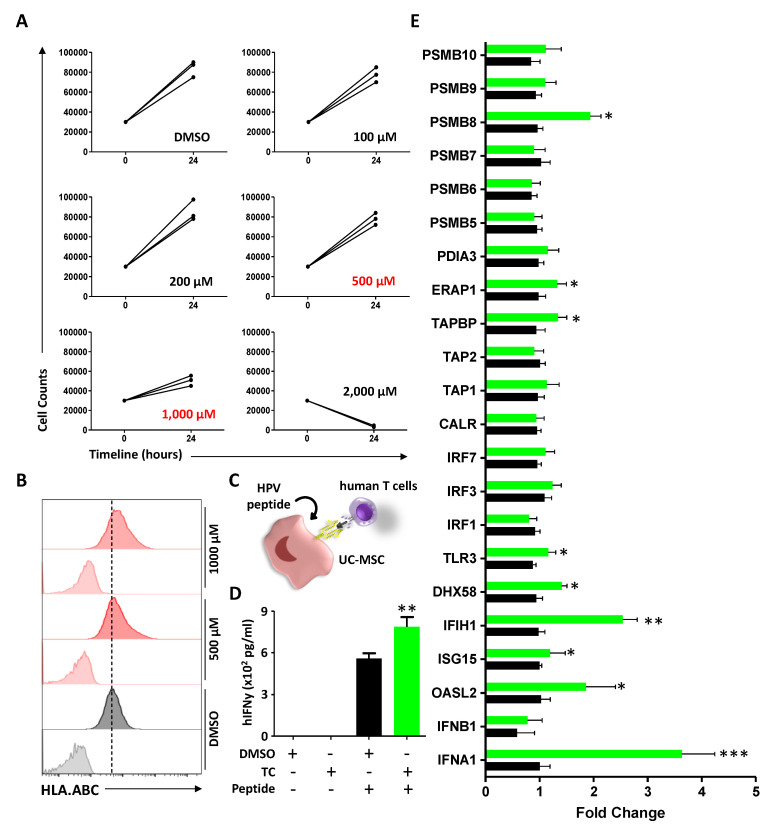
TC treatment leads to similar outcomes on human UC-MSCs. (**A**) Identification of the MTD for TC on human UC-MSCs. (**B**) Representative flow-cytometry analysis of HLA-ABC expression following DMSO (gray) or TC (red) treatments. (**C**) Schematic diagram of the antigen presentation assay using UC-MSCs. (**D**) Antigen presentation assay using TC-treated human UC-MSCs stimulated with the YMLDLQPET HPV peptide. (**E**) Quantification of gene transcripts in response to TC treatment. For this panel, n = 5/group with * *p* < 0.05, ** *p* < 0.01, and *** *p* < 0.001.

## Data Availability

Not applicable.

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
