# Peer review of "LSD1 Inhibition Enhances the Immunogenicity of Mesenchymal Stromal Cells by Eliciting a dsRNA Stress Response"

_cells, 2022, doi:10.3390/cells11111816_

Round 1

Reviewer 1 Report

The manuscript entitled “LSD1 Inhibition enhances the immunogenicity of mesenchymal stromal cells by eliciting a dsRNA stress response” revealed another pharmacological strategy to enhance MSC immunogenicity. They found inhibition of LSD1 can covert MSC to APC-like cells, which. Overall, the project was well designed. However, there were some issues need to be solved:

Major comments:

  1. In the introduction section, the authors should clearly put forward scientific questions and summarize the major results and conclusions derived from this paper.
  2. The authors use TC to inhibit LSD1, but there are no data can verify LSD1 is successfully inhibited by TC at 500 um concentration. WB and IF should be used to analysis LSD1 and H3K4me1/2 expression after TC treatment.
  3. To prove LSD1 mediated the dsRNA sensing and MSC antigen presentation, rescue experiment should be provided, to see if re-expressing LSD1 in TC treated MSC could neutralize IFNbeta expression and antigen-presentation ability of MSC.
  4. It should be interesting to compare the anti-tumor ability of TC- and UM171a-treated MSCs, since this result could further highlight the importance of this study.

Minor comments (there are many spelling mistakes and annotation problems, the authors should take them seriously):

  1. Line 44, “speficic” should be “specific”.
  2. Line 51, “Althouhg” should be “Although”
  3. Line 59, “estabslish” should be “establish”
  4. Line 437, “demonstre” should be “demonstrate”
  5. Figure 1B, the authors should annotate red and gray group in the figure
  6. Figure 2A, the setSize on the right isn’t match the dot size in the dot plot
  7. Figure 3B, the two groups should be “DMSO” and “TC”

Author Response

REVIEWER 1

Comment 1

In the introduction section, the authors should clearly put forward scientific questions and summarize the major results and conclusions derived from this paper.

Response to comment 1

We appreciate the comment of reviewer 1 in that regard. We have thus added the following section at the end of the introduction, which in addition to the discussion, would better clarify our rationale, goals and findings:

"Such study will not only unveil whether new pharmacological compounds can convert MSCs into APCs, but it will also provide fundamental knowledge regarding the mode of action of these two compounds with respect to targeting the entire CoREST complex (for UM171a) as opposed to a specific component within this complex (specific inhibitor of LSD1).  In a nutshell, this is the first study demonstrating that tranylcypromine (TC)-mediated LSD1 inhibition in MSCs induces UM171a-distinct APC-like functions through a dsRNA stress response consequently resulting in efficient and stable peptide presentation to responding CD8 T cells and potent anti-tumoral effector responses".   

Comment 2

The authors use TC to inhibit LSD1, but there are no data can verify LSD1 is successfully inhibited by TC at 500 um concentration. WB and IF should be used to analysis LSD1 and H3K4me1/2 expression after TC treatment.

Response to comment 2

We agree with the reviewer that additional experiments could be conducted to re-enforce the notion of successful LSD1 inhibition using the TC compound. However, we believe that this is beyond the scope of the current study. We have conducted a series of analyses based on a previously published study (Sheng W. et al. Cell 2018) to demonstrate TC-mediated effects in treated MSCs. For instance, our data replicate what have been observed by others with regards to dsRNA induction (Fig. 3A-B) as well as the expression of various genes associated with a TC response including Type-I IFNs, PRR and IRFs (Fig. C-G). We thus believe that these observations, which correlate perfectly with an LSD1 inhibition response, reflect a direct effect mediated by TC.

Comment 3

To prove LSD1 mediated the dsRNA sensing and MSC antigen presentation, rescue experiment should be provided, to see if re-expressing LSD1 in TC treated MSC could neutralize IFNbeta expression and antigen-presentation ability of MSC.

Response to comment 3

We thank the reviewer for this suggestion. Please refer to the response provided above (to comment 2).

Comment 4

It should be interesting to compare the anti-tumor ability of TC- and UM171a-treated MSCs, since this result could further highlight the importance of this study.

Response to comment 4

We thank the reviewer for this interesting suggestion. In fact, the initial objective was to compare the therapeutic efficacy of TC- versus UM171a-treated MSCs in the context of therapeutic vaccination. However, we changed our initial objective as TC treatment did not lead to antigen cross-presentation as did UM171a. In other words, TC and UM171a did not affect MSCs in an equivalent way. TC triggered enhanced antigen presentation as opposed to UM171a, which instilled antigen cross-presentation (our main objective). Since both treatments affect the immune-related function of MSCs in a divergent way, we decided to solely test TC-treated MSCs as a cell vaccine pulsed with a given peptide.

Comment 5

Minor comments (there are many spelling mistakes and annotation problems, the authors should take them seriously):

  1. Line 44, “speficic” should be “specific”.
  2. Line 51, “Althouhg” should be “Although”
  3. Line 59, “estabslish” should be “establish”
  4. Line 437, “demonstre” should be “demonstrate”
  5. Figure 1B, the authors should annotate red and gray group in the figure
  6. Figure 2A, the setSize on the right isn’t match the dot size in the dot plot
  7. Figure 3B, the two groups should be “DMSO” and “TC”

Response to comment 5

We thank the reviewer for pointing-out these typos and minor mistakes. All corrections have been implemented as requested.

Reviewer 2 Report

In the paper "LSD1 Inhibition enhances the immunogenicity of mesenchymal stromal cells by eliciting a dsRNA stress response" the authors present a new way of inducing APC properties of MSCs in order to improve anti-tumoral immunity, and demonstrate some mechanistic explanations of this effect.

The manuscript is very well written. The study is presented in a clear and comprehensive way. It is well designed to appropriately assess the hypothesis. The methodology is described in detail so that the results could be reproduced. The data are presented clearly and fully, and are interpreted properly. The conclusion is supported by the data.

The one thing the authors could improve in the introduction is to better explain the need and the advantages of direct inhibition of LSD1 to the use of UM171a as a therapeutic. Also, I would suggest the authors to repeat full names of some abbreviations in the main text that are introduced in the abstract (e.g. LSD1, TC), in order to enhance reading fluency.

Overall, the manuscript is interesting, scientifically sound, and gives clues to new ways to improve cell therapies to fight cancer.

Author Response

REVIEWER 2

Comment 1

The one thing the authors could improve in the introduction is to better explain the need and the advantages of direct inhibition of LSD1 to the use of UM171a as a therapeutic.

Response to comment 1

We appreciate the comment of reviewer 2 in that regard. We have thus added the following section at the end of the introduction, which in addition to the discussion, would better clarify our rationale, goals and findings:

"Such study will not only unveil whether new pharmacological compounds can convert MSCs into APCs, but it will also provide fundamental knowledge regarding the mode of action of these two compounds with respect to targeting the entire CoREST complex (for UM171a) as opposed to a specific component within this complex (specific inhibitor of LSD1).  In a nutshell, this is the first study demonstrating that tranylcypromine (TC)-mediated LSD1 inhibition in MSCs induces UM171a-distinct APC-like functions through a dsRNA stress response consequently resulting in efficient and stable peptide presentation to responding CD8 T cells and potent anti-tumoral effector responses."   

Comment 2

Also, I would suggest the authors to repeat full names of some abbreviations in the main text that are introduced in the abstract (e.g. LSD1, TC), in order to enhance reading fluency.

Response to comment 2

Changes have been implemented as requested by reviewer 2 (mainly in the introduction).

Reviewer 3 Report

This is a very well-written manuscript  demonstrating that the treatment of MSCs with the LSD1 inhibitor tranylcypromine (TC) elicits a double-stranded (ds)RNA stress response along with its associated responsive elements including pattern recognition receptors (PRRs), Type-I interferon (IFN) and IFN-stimulated genes (ISGs). According to the authors, the net outcome culminates in enhanced expression of H2-Kb, and increased stability of the cell surface peptide:MHCI complexes. As a result, TC-treated MSCs stimulate CD8 T-cell activation efficiently and elicit potent anti-tumoral responses against the EG.7 T-cell lymphoma. The authors conclude that these findings reveal a new pharmacological protocol whereby targeting LSD1 in MSCs elicits APC-like capabilities that could be easily exploited in the design of future MSC-based anti-cancer vaccines.

What I miss in this manuscript is more informations on human UC-derived MSCs:

  • Do the not-treated human UC-MSC with TC express HLA-Class I antigens (-A, -B and -C) and how much (control group)
  • How much of them express these antigens (in %) after the treatment with TC and 
  • Do the treated UC-MSC with TC express also costimulatory molecules CD80/CD86, which play an important role in antigen presentation.
  • Did the authors perform this kind of experiments with bone marrow-derived MSCs?

Author Response

REVIEWER 3

Comment 1

Do the not-treated human UC-MSC with TC express HLA-Class I antigens (-A, -B and -C) and how much (control group)? How much of them express these antigens (in %) after the treatment with TC?

Response to comment 1

In fact, all UC-MSCs, treated or not, express high levels of HLA-ABC as shown in Figure 7B. In other words, all cells are 100% positive. This is why a red dotted line was added to demonstrate difference in the intensities of histogram peeks (e.g. different mean fluorescent intensities).

Comment 3

Do the treated UC-MSC with TC express also costimulatory molecules CD80/CD86, which play an important role in antigen presentation?

Response to comment 3

This is indeed a very important point. We did look at CD80 and CD86 expression on the surface of both DMSO- and TC-treated MSCs and found no detectable signal for both of these co-stimulatory markers. If we assume that murine and/or human MSCs in general do not express the co-stimulatory signal, then how can they mount an immune response once delivered in vivo? The answer to this question is rather simple. MSCs are know nowadays to undergo efferocytosis following their in vivo delivery (whether there is inflammation or not – Galleu A. et al. Sci Transl Med. 2017; Pang SHM et al. Nat Commun 2021). As such, injected MSCs are captured by resident phagocytes and DCs (a way of cross-priming endogenous APCs) consequently resulting in syngeneic TCR:MHC interaction resulting in T-cell activation and anti-tumoral responses. We are currently conducting studies to decipher the phagocytic cell population phenotype and required activation signals, which we can then exploit to enhance in vivo-mediated efferocytosis in response to MSC-mediated vaccination.

Comment 4

Did the authors perform this kind of experiments with bone marrow-derived MSCs?

Response to comment 4

No, we focused on the human MSCs that were obtained from our commercial partner for this proof-of-concept. We are however working with an industrial partner on the development of another MSC-based anti-cancer vaccine testing UC-, BM- and adipose-derived MSCs as a means to identify the "best-acting" MSC population.

Round 2

Reviewer 1 Report

The authors have addressed all my concerns.